# GStarX: Explaining Graph Neural Networks with Structure-Aware Cooperative Games

**Shichang Zhang**[1]    **Yozen Liu**[2]    **Neil Shah**[2]    **Yizhou Sun**[1]
[1]University of California, Los Angeles    [2]Snap Inc.
[1]{shichang, yzsun}@cs.ucla.edu    [2]{yliu2, nshah}@snap.com

## Abstract

Explaining machine learning models is an important and increasingly popular area of research interest. The Shapley value from game theory has been proposed as a prime approach to compute feature importance towards model predictions on images, text, tabular data, and recently graph neural networks (GNNs) on graphs. In this work, we revisit the appropriateness of the Shapley value for GNN explanation, where the task is to identify the most important subgraph and constituent nodes for GNN predictions. We claim that the Shapley value is a non-ideal choice for graph data because it is by definition not structure-aware. We propose a G̲raph S̲tructure-aw̲are eX̲planation (GStarX) method to leverage the critical graph structure information to improve the explanation. Specifically, we define a scoring function based on a new structure-aware value from cooperative game theory proposed by Hamiache and Navarro (HN). When used to score node importance, the HN value utilizes graph structures to attribute cooperation surplus between neighbor nodes, resembling message passing in GNNs, so that node importance scores reflect not only the node feature importance, but also the node structural roles. We demonstrate that GStarX produces qualitatively more intuitive explanations, and quantitatively improves explanation fidelity over strong baselines on chemical graph property prediction and text graph sentiment classification.[1]

## 1   Introduction

Explainability is crucial for complex machine learning (ML) models in sensitive applications, helping establish user trust and providing insights for potential model improvements. Many efforts focus on explaining models on images, text, and tabular data. In contrast, the explainability of models on graph data is yet underexplored. Since explainability can be especially critical for many graph tasks like drug discovery, and interest in deep graph models is growing rapidly, further investigation of graph explainability is warranted. In this work, we study graph ML explanation with graph neural networks (GNNs) as the target models, given their popularity and widespread use for graph machine learning tasks [42, 29, 38, 34, 33, 45].

In ML explainability, important features are identified, and the Shapley value [30] has been deemed as a "fair" scoring function for computing feature importance. Originally from cooperative game theory, many values, including the Shapley value, have been proposed for allocating a total payoff to players in a game. When used for scoring the feature importance of a data instance, the model prediction is treated as the total payoff and the features are considered as players. In particular, for an instance with $n$ features $\{\boldsymbol{x}_1, \ldots \boldsymbol{x}_n\}$, the Shapley value of its $i$th feature $\boldsymbol{x}_i$ is computed via aggregating $m(i, S)$, which are the marginal contributions of $\boldsymbol{x}_i$ to sets of other features $\boldsymbol{x}_S \subseteq \{\boldsymbol{x}_1, \ldots, \boldsymbol{x}_n\} \setminus \{\boldsymbol{x}_i\}$. Each $\boldsymbol{x}_S$ is called a *coalition*. Each $m(i, S)$ is computed as the difference between model outputs for

---

[1]Code available at https://github.com/ShichangZh/GStarX

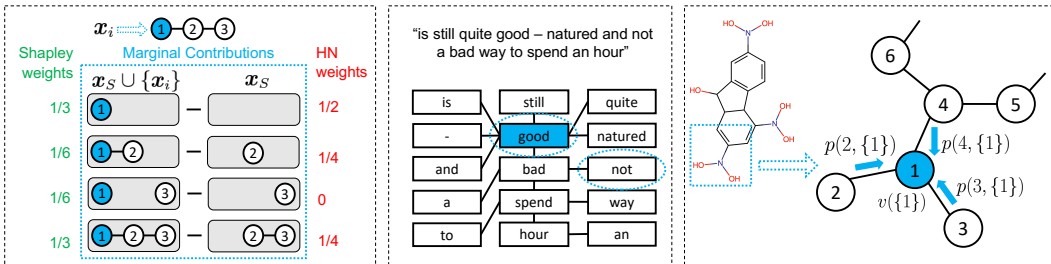

Figure 1: Explanations on graphs with structure-aware values (like HN) offers advantages over non-structure-aware values (like Shapley). **(a) Synthetic graph (left):** The Shapley value assigns weights to $m(i, S)$ only based on size of $\boldsymbol{x}_S$, while the HN value assigns weights considering structures and in particular gives zero weight to the disconnected $\boldsymbol{x}_S$. **(b) Text graph (middle):** For a sentence classified as positive, the {"not", "good"} coalition shouldn't be considered when they are not connected by "bad". **(c) Chemical graph (right):** For a chemical graph with mutagenic functional group -NO2, the importance of the atom N (node 1) is better recognized if decided locally within the functional group.

$\boldsymbol{x}_S \cup \{\boldsymbol{x}_i\}$ and $\boldsymbol{x}_S$, e.g., difference of probability belonging to a target class for these two set of features, and it is meant to capture the interaction between $\boldsymbol{x}_i$ and $\boldsymbol{x}_S$. The Shapley value is widely used for explaining ML models on images, text, and tabular data, when the features are pixels, words, and attributes [22, 24].

The Shapley value has recently been extended to explain GNNs on graphs through feature importance scoring as above, where features are nodes [9] or supernodes [44]. We argue that the Shapley value is a non-ideal choice for (super)node importance scoring because its contribution aggregation is **non-structure-aware**. The Shapley value aggregation assumes no structural relationship between $\boldsymbol{x}_i$ and $\boldsymbol{x}_S$ even though they are both parts of the input graph (a review of the Shapley value is in Section 2.2). Since the graph structure generally contains critical information and is crucial to the success of GNNs, we consider properly leveraging the structure with a better **structure-aware** scoring function.

We propose Graph Structure-aware eXplanation (GStarX), where we construct a structure-aware node importance scoring function based on the Hamiache-Navarro (HN) value [15] from cooperative game theory. Recall that GNNs make predictions via message passing, during which node representations are learned by aggregating messages from neighbors. Message passing aggregates both feature and structure information, resulting in powerful structure-aware models [5]. The HN value shares a similar idea to message passing by allocating the payoff surplus generated from the cooperation between neighboring players (nodes). When used as a scoring function to explain node importance, the HN value captures both features and structural interactions between nodes (details in Section 4). Figure 1(**a**) shows an example comparing the Shapley value and the HN value. In this example, their difference boils down to different aggregation weights of marginal contributions, where the former is uniform and the latter is structure-aware (details in Section 3.2). In summary, our contributions are:

- Identify the non-structure-aware limitation of the Shapley value for GNN explanation.
- Introduce the structure-aware HN value from cooperative game theory to the graph machine learning community and connect it to the GNN message passing and GNN explanation.
- Propose a new HN-value-based GNN explanation method GStarX, and demonstrate the superiority of GStarX over strong baselines for explaining GNNs on chemical and text graphs.

## 2 Preliminaries

### 2.1 Graph neural networks

Consider a graph $\mathcal{G}$ with (feature-enriched) nodes $\mathcal{V}$ and edges $\mathcal{E}$. We denote $\mathcal{G}$ as $\mathcal{G} = (\mathcal{V}, \boldsymbol{X}, \boldsymbol{A})$, where $\boldsymbol{X} \in \mathbb{R}^{n \times d}$ denotes $d$-dimensional features of $n$ nodes in $\mathcal{V}$, and $\boldsymbol{A} \in \{0, 1\}^{n \times n}$ denotes the adjacency matrix specifying edges in $\mathcal{E}$. GNNs make predictions on $\mathcal{G}$ by learning representations via the *message-passing* mechanism. During message passing, the representation of each node $u \in \mathcal{V}$ is updated by aggregating its own representation and representations (messages) from its neighbors. We denote the set of neighbors as $\mathcal{N}(u)$. This aggregation is recursively applied, so $u$ can collect

messages from its multi-hop neighbors and produce structure-aware representations [5]. With $\boldsymbol{h}_i^{(l)}$ denotes the representation of node $i$ at iteration $l$, and $\text{AGGR}(\cdot, \cdot)$ denotes the aggregation operation, e.g. summation, the representation update is shown in Equation 1.

$$\boldsymbol{h}_u^{(l)} = \text{AGGR}(\boldsymbol{h}_u^{(l-1)}, \{\boldsymbol{h}_i^{(l-1)} | i \in \mathcal{N}(u)\}) \tag{1}$$

## 2.2 Cooperative games

**A cooperative game** denoted by $(N, v)$, is defined by a set of players $N = \{1, \ldots, n\}$, and a *characteristic function* $v : 2^N \to \mathbb{R}$. $v$ takes a subset of players $S \subseteq N$, called a *coalition*, and maps it to a payoff $v(S)$, where $v(\emptyset) := 0$. A *solution function* $\phi$ is a function maps each given game $(N, v)$ to $\phi(N, v) \in \mathbb{R}^n$. The vector $\phi(N, v)$, called a *solution*, represents a certain allocation of the total payoff $v(N)$ generated by all players to each individual, with the $i$th coordinate $\phi_i(N, v)$ being the payoff attributed to player $i$. $\phi(N, v)$ is also called the "value" of the game when it satisfies certain properties, and different values were proposed to name solutions with different properties [30, 35].

**The Shapley value** is one popular solution of cooperative games. The main idea is to assign each player a "fair" share of the total payoff by considering all possible player interactions. For example, when player $i$ cooperates with a coalition $S$, the total payoff $v(S \cup \{i\})$ may be very different from $v(S) + v(\{i\})$ because of $i$'s interaction with S. Thus the marginal contribution of $i$ to $S$ is defined as by $m(i, S) = v(S \cup \{i\}) - v(S)$. Then the formula of the Shapley value for $i$ is shown in Equation 2, where marginal contributions to all possible coalitions $S \subseteq N \backslash \{i\}$ are aggregated. The first identify in Equation 2 shows that the aggregation weights are first uniformly distributed among coalition sizes $k$ (outer average), then uniformly distributed among all coalitions with the same size (inner average).

$$\phi_i(N, v) = \overbrace{\frac{1}{n} \sum_{k=0}^{n-1}}^{\text{Average over } k} \overbrace{\frac{1}{\binom{n-1}{k}} \sum_{\substack{S \subseteq N \backslash \{i\} \\ |S|=k}}}^{\text{Average over } S \text{ s.t. } |S|=k} m(i, S) = \sum_{S \subseteq N \backslash \{i\}} \frac{|S|!(n - |S| - 1)!}{n!} m(i, S) \tag{2}$$

**Games with communication structures.** Although the Shapley value is widely used for cooperative games, its assumption of fully flexible cooperation among all players may not be achievable. Some coalitions may be preferred over others and some may even be impossible due to limited communication among players. Thus, [26] uses a graph $\mathcal{G}$ as the *communication structure* of players to represent cooperation preference. A game with a communication structure is defined by a triple $(N, v, \mathcal{G})$, with $N$ being the node set of $\mathcal{G}$. This game formulation is more practical than fully flexible cooperation when cooperation preference is available. Several values with different properties have been proposed for such games [26, 2, 13, 18] including the HN value [15].

# 3 GNN explanation via feature importance scoring

## 3.1 Problem formalization

A general approach to formalize an ML explanation problem is through feature importance scoring [24, 6], where features may refer to pixels of images, words of text, or nodes/edges/subgraphs of graphs. Let $f(\cdot)$ denote a to-be-explained GNN, $\mathcal{G} = (\mathcal{V}, \boldsymbol{X}, \boldsymbol{A})$ denote an input graph, and $0 < \gamma < 1$ denote a sparsity constraint to enforce concise explanation. GNN explanation via subgraph scoring is aimed to find a subgraph $g$ that maximizes a given evaluation metric $\text{EVAL}(\cdot, \cdot, \cdot)$, which measures the faithfulness of $g$ to $\mathcal{G}$ regarding making predictions with $f(\cdot)$, i.e.

$$g^* = \underset{g \subseteq \mathcal{G}, |g| \leq \gamma |\mathcal{G}|}{\arg \max} \ \text{EVAL}(f(\cdot), \mathcal{G}, g) \tag{3}$$

When the task is graph classification and $f(\cdot)$ outputs a one-sum vector $f(\mathcal{G}) \in [0, 1]^C$ containing probabilities for $\mathcal{G}$ belongs to $C$ classes, an example EVAL can be the prediction probability drop for removing $g$ from $\mathcal{G}$, i.e. $\text{EVAL}(f(\cdot), \mathcal{G}, g) = [f(\mathcal{G})]_{c^*} - [f(\mathcal{G} \backslash g)]_{c^*}$ with $c^* = \arg \max_c [f(\mathcal{G})]_c$.

In practice, since the number of subgraphs is combinatorial in the number of nodes, the objective is often relaxed to finding a set of important nodes or edges first and then inducing the subgraph

[41, 25, 9]. A more tractable objective of finding the optimal set of nodes $S^* \subseteq \mathcal{V}$ [2] is given by

$$S^* = \underset{S \subseteq \mathcal{V}, |S| \leq \gamma |\mathcal{V}|}{\arg\max} \sum_{i \in S} \text{SCORE}(f(\cdot), \mathcal{G}, i) \qquad (4)$$

Existing methods often boil down to Equation 4 with different scoring functions (SCORE), and finding a proper SCORE is non-trivial. One example of SCORE is to evaluate each node $i$ directly as $\text{SCORE}(f(\cdot), \mathcal{G}, i) = [f(\{i\})]_{c^*}$. However, this choice misses interactions between nodes and corresponds to a trivial case in GNNs where no message-passing is performed for $\{i\}$. Another possibility is to use EVAL as SCORE, e.g., $\text{SCORE}(f(\cdot), \mathcal{G}, i) = [f(\mathcal{G})]_{c^*} - [f(\mathcal{G} \backslash \{i\})]_{c^*}$. However, this again fails to capture interactions between nodes; for example, two nodes $i$ and $j$ may be both important but also complimentary, so their contribution to $\mathcal{G}$ can only be observed when they are missing simultaneously.

## 3.2 Scoring functions from cooperative games

Given the challenges for defining a proper SCORE, solutions to cooperative games, like the Shapley value, have been proposed with $f(\cdot)$ as the characteristic function, i.e. $\text{SCORE}(f(\cdot), \mathcal{G}, i) = \phi_i(|\mathcal{G}|, f(\cdot))$ [44, 9]. However, existing works only use the non-structure-aware Shapley value. In contrast, values defined on games $(N, v, \mathcal{G})$ with communication structures $\mathcal{G}$ are naturally structure-aware but were never considered GNN explanation. Below we discuss the non-structure-aware limitation of the Shapley value in detail and motivating structure-aware values with practical examples in GNN explanation.

The Shapley value is defined on games $(N, v)$, which by definition takes no graph structures. It assumes flexible cooperation between players and uniform distribution of coalition importance that only depends on $|S|$ (see Equation 2). Even if a $\mathcal{G}$ is given and the game is defined as $(N, v, \mathcal{G})$, the Shapley value will overlook $\mathcal{G}$ when aggregating $m(i, S)$. In contrast, structure-aware values on $(N, v, \mathcal{G})$ can be interpreted as a weighted aggregation of coalitions with more reasonable weights. Although different solutions $\phi(N, v, \mathcal{G})$ have their nuances in weight adjustments [13, 15, 26, 18], they share two key properties: **(1)** the weight is zero if $i$ and $S$ are disconnected because they are interpreted as players without communication channels [26], and **(2)** the weight is impacted by the nature of connections between $i$ and $S$ because it is easier for better-connected nodes to communicate.

**A synthetic example.** We take the HN value (definition in Section 4.1) as an example structure-aware value and compare it to the Shapley value in a simple graph in Figure 1**(a)**. To compute $\phi_1(N, v, \mathcal{G})$, both values aggregates $m(1, S)$ for $S \in \{\emptyset, \{2\}, \{3\}, \{2, 3\}\}$. The Shapley value first assigns a uniform weight $\frac{1}{3}$ to three different $|S|$, and then splits weights uniformly for the $|S| = 1$ case to be $\frac{1}{6}$. However, the HN value assigns weight zero for $S = \{3\}$ because 1 and 3 are disconnected in coalition $\{1, 3\}$ and are assumed to be two independent graphs that shouldn't interact (property **(1)**). Their interaction is rather captured in the $S = \{2, 3\}$ case, when 1 and 3 are connected by the bridging node 2, and this case is also downweighted from $\frac{1}{3}$ to $\frac{1}{4}$, as 3 is relatively far from 1 (property **(2)**).

**A practical example.** The good properties of structure-aware values can help explain graph tasks. The example in Figure 1**(b)** is from `GraphSST2` (dataset description in Section 5.1), where the graph for sentiment classification is constructed from the sentence *"is still quite good-natured and not a bad way to spend an hour"* with edges generated by the Biaffine parser [12]. Assuming a model can correctly classify it as positive. Intuitively, "good" and "not a bad" are central to the human explanation. To compute the Shapley value of the word "good", the coalition "not good" will diminish the positive importance of "good", despite the two words lacking any direct connection. A structure-aware value can instead eliminate the {"not", "good"} coalition, and only consider interactions between "not" and "good" (in fact, "not" and any other word) when the bridging "bad" appears, hence better binding "not" with "bad" and improving the salience of "good". In Section 5.2, we revisit this example to observe impacts of structure-awareness empirically.

---

[2] A similar objective can be defined as $S$ over edges $\mathcal{E}$. We define it over nodes as nodes often contain richer features than edges and are more flexible. One advantage of this choice will be made clear in Section 5.2

# 4 GStarX: Graph Structure-aware eXplanation

We propose GStarX, which uses a structure-aware HN-value-based SCORE to explain GNNs. We first state the definition of the HN value in cooperative game theory (4.1), and then connect it to the GNN message passing (4.2), and finally give the GStarX algorithm for GNN explanation (4.3).

## 4.1 The HN value

Let $(N, v, \mathcal{G})$ be a game with a communication structure $\mathcal{G}$ and $S \subseteq N$ be a coalition. Let $\bar{S} = \cup_{i \in S}\{\mathcal{N}(i)\} \cup S$ to be the union of $S$ and its neighbors in $\mathcal{G}$. Let $S/\mathcal{G}$ be the partition of $S$ containing connected components in $\mathcal{G}$, i.e., $S/\mathcal{G} = \{\{i | i = j \text{ or } i \text{ and } j \text{ are connected in } S \text{ by } \mathcal{E} \text{ of } \mathcal{G}\} | j \in S\}$. Let $\mathcal{G}[S]$ be the induced subgraph of $S$ in $\mathcal{G}$. For example, in Figure 1(**b**), when $S =$ {"is", "an", "hour"}, $\bar{S}$ will be {"is", "good", "an", "hour", "spend"}, $S/\mathcal{G}$ will be {{"is"}, {"an", "hour"}}, and $\mathcal{G}[S]$ will be the subgraph with a two-node component $\boxed{\text{an}}-\boxed{\text{hour}}$ and a single node component $\boxed{\text{is}}$.

**Definition 4.1** (**Surplus**). The surplus $p(j, S)$ generated by a coalition $S$ cooperating with its neighbor $j$ is defined as

$$p(j, S) = v(S \cup \{j\}) - v(S) - v(\{j\}) \tag{5}$$

Intuitively, $p(j, S)$ is generated because $S$ is actively cooperating. Thus, when evaluating a fair payoff to $S$, a portion of $p(j, S)$ should be added to its own payoff $v(S)$. This idea leads to the next definition of associated games regarding the original games, where surplus allocation is performed.

**Definition 4.2** (**HN Associated Game**). Given $0 \leq \tau \leq 1$ representing the portion of surplus that will be allocated to a coalition $S$ for its cooperation with other players. The HN associated game $(N, v_\tau^*, \mathcal{G})$ of $(N, v, \mathcal{G})$ is defined as

$$v_\tau^*(S) = \begin{cases} v(S) + \tau \sum_{j \in \bar{S} \setminus S} p(j, S) & \text{if } |S/\mathcal{G}| = 1 & (6) \\ \sum_{T \in S/\mathcal{G}} v_\tau^*(T) & \text{otherwise} & (7) \end{cases}$$

The HN value is a solution on $(N, v, \mathcal{G})$. It is computed by iteratively constructing a series of HN associated games until it converges to a *limit game* $(N, \tilde{v}, \mathcal{G})$. In other words, we first construct $v_\tau^*$ from $v$ by surplus allocation. Then we construct $v_\tau^{**}$ from $v_\tau^*$ by allocating the surplus generated from the $v_\tau^*$ and so on. The convergence of the limit game is guaranteed and the result $\tilde{v}$ is independent of $\tau$ under mild conditions as shown in [15]. The HN value of each player is uniquely determined by applying $\tilde{v}$ to that player, i.e. $\phi_i(N, v, \mathcal{G}) = \tilde{v}(\{i\})$. We state the formal definitions of the limit game and the uniqueness theorem of the HN value in Appendix E.2.

## 4.2 Connecting GNNs and the HN surplus allocation through the message passing lens

Both the GNN message passing (MP) and the associated game surplus allocation (SA) are iterative aggregation algorithms, with considerable alignment. In fact, SA on each singular node set $S = \{i\}$ is exactly MP: Equation 6 becomes an instantiation of Equation 1 with $\text{AGGR}(a, \boldsymbol{b}) = a + \tau \sum_j \boldsymbol{b}_j$ on a scalar node value $a$ and a neighbor set $\boldsymbol{b}$. These algorithms differ in that SA applies more broadly to $|S| \geq 1$ cases; it treats $S$ as a supernode when nodes in $S$ form a connected component in $\mathcal{G}$, and handles disconnected $S$ component-wise via Equation 7.

We illustrate SA using a real chemical graph example. The molecule shown in Figure 1(**c**) is taken from MUTAG (dataset description in Section 5.1). It is known to be classified as *mutagenic* because of the -NO2 group (nodes 1, 2, and 3) [8]. When we compute $v_\tau^*(\{1\})$, the surplus $p(2, \{1\})$, $p(3, \{1\})$, and $p(4, \{1\})$ are allocated to node 1 (like messages passed to a central node in GNN). Then surplus are aggregated together with $v(\{1\})$ following Equation 6 to form $v_\tau^*(\{1\})$.

For graphs, the SA approach has two advantages over the uniform aggregation approach used in the Shapley value: (**1**) The aggregated payoff in each $v_\tau^*$ is structure-aware, like representations learned by GNNs [5], and (**2**) the iterative computation preserves locality, which is preserved by GNNs [3]. In other words, these two properties mean close neighbors heavily influence each other due to cooperation in many iterations, while far away nodes less influence each other due to little

| **Algorithm 1** GStarX: Graph Structure-Aware Explanation | **Algorithm 2** The Compute-HN Function |
|---|---|

**Input:** Graph $\mathcal{G}$ with nodes $\mathcal{V} = \{u_1, \ldots, u_n\}$, trained GNN $f(\cdot)$, empirical expectation $f^0$, hyperparameter $\tau$, max sample size $m$, number of samples $J$, sparsity $\gamma$.
Get the predicted class $c^* = \arg\max_c [f(\mathcal{G})]_c$
Define characteristic function $v(S) = [f(g_S)]_{c^*} - f^0_{c^*}$
**if** $n \leq m$ **then**
    $\phi = $ Compute-HN$(\mathcal{G}, \mathcal{V}, v(\cdot), \tau)$
**else**
    $\phi = $ Compute-HN-MC$(\mathcal{G}, \mathcal{V}, v(\cdot), \tau, m, J)$
**end if**
Sort $\phi$ in descending order with indices $\{\pi_1, \ldots, \pi_n\}$
$k = \lfloor \gamma |\mathcal{V}| \rfloor$
**Return:** $S^* = \{u_{\pi_1}, \ldots, u_{\pi_k}\}$

**Input:** Graph instance $\mathcal{G}$ with nodes $\mathcal{V} = \{u_1, \ldots, u_n\}$, characteristic function $v$, hyperparameter $\tau$.
**for** $S$ in $2^N$ **do**
    Compute payoff $v(S)$ {Eq.(8)}
**end for**
Construct matrix $\boldsymbol{H}_{\{\tau, n, \mathcal{G}\}}$ {Eq.(16)}
**repeat**
    $\boldsymbol{H} = \boldsymbol{H}\boldsymbol{H}$
**until** $\boldsymbol{H}$ converges
Get the limit game $\tilde{v} = \boldsymbol{H}v$ {Eq.(17)}
Assign the first $n$ entries of $\tilde{v}$ to $\phi$
**Return:** $\phi$

cooperation. In the MUTAG example, since the local -NO2 generates a high payoff for the mutagenicity classification, locally allocating the payoff helps us better understand the importance of the nitrogen atom and the oxygen atoms. Whereas aggregating over many unnecessary coalitions with far-away carbon atoms can obscure the true contribution of -NO2. We will revisit this example in Section 5.2.

## 4.3 The GStarX algorithm

We now state our algorithm for explaining GNNs with GStarX. Notice that GStarX scores nodes in a graph but not each dimension of node features. Feature dimension importance explanation is an orthogonal perspective that can be added on top of GStarX. We leave this extension as a future work. GStarX formulates the GNN explanation problem as a feature importance scoring problem, where nodes are scored to find the optimal node-induced subgraph as we introduced in Section 3.1. It essentially implements and solves the objective in Equation 4, where an HN-value-based SCORE is used. To use such SCORE, we need to define the players and the characteristic function of the game, and then apply the formula in Equation 6 and 7. Suppose the inputs are a graph $\mathcal{G}$ with nodes $\mathcal{V} = \{u_1, \ldots, u_n\}$ and label $y \in \{1, \ldots, C\}$, a GNN $f(\cdot)$ outputs a probability vector $f(\mathcal{G}) \in [0, 1]^C$, and the predicted class $c^* = \arg\max_c [f(\mathcal{G})]_c$. Let $\mathcal{V}$ be players, and let the normalized probability of the predicted class be the characteristic function $v$:

$$v(S) = [f(\mathcal{G}[S])]_{c^*} - f^0_{c^*} \quad \forall S \subseteq \mathcal{V} \tag{8}$$

Here the normalization term $f^0_{c^*} = \mathbb{E}\left[[f(G)]_{c^*}\right]$ is the expectation over a random variable $G$ representing a general graph. In practice, we approximate it using the empirical expectation over all $\mathcal{G}$ in the dataset. SCORE will be the HN value of the game, i.e., SCORE$(f(\cdot), \mathcal{G}, i) = \phi_i(\mathcal{V}, v, \mathcal{G}) = \tilde{v}(\{i\})$.

Given SCORE, we solve the objective by first computing the scores $\phi \in \mathbb{R}^n$ then selecting the top $\lfloor \gamma |\mathcal{V}| \rfloor$ scores greedily as in Algorithm 1. Practically, like other game-theoretic methods, the exact computation of the HN value is infeasible when the number of players $n$ is large. We thus do an exact computation for small graphs (the if-branch) and Monte-Carlo sampling for large graphs (the else-branch). The Compute-HN function is shown in Algorithm 2, where the $\boldsymbol{H}$ stands for a matrix form of the associated game defined in Definition 4.2.(See Appendix E.2 and E.3 for details of the matrix form and algorithms for Compute-HN-MC). Also, even though the algorithm is stated for graph classification, GStarX works for node classification as well. This can be easily seen since GNNs classify nodes $u_i$ by processing an ego-graph centered at $u_i$, so the task can be converted to graph classification with the label of $u_i$ used as the label of the ego-graph. We focus on graph classification in the main text for simpler illustration and discuss more about node classification in Appendix B.

# 5 Experiments

## 5.1 Experiment settings

**Datasets.** We conduct experiments on datasets from different domains including synthetic graphs, chemical graphs, and text graphs. A brief description of the datasets is shown below with more detailed statistics in Appendix A.1

- **Chemical graph property prediction.** `MUTAG` [8], `BACE` and `BBBP` [39] contain chemical molecule graphs for graph classification, with atoms as nodes, bonds as edges, and chemical properties as graph labels.
- **Text graph sentiment classification.** `GraphSST2` and `Twitter` [43] contain graphs constructed from text. Nodes are words with pre-trained BERT embeddings as features. Edges are generated by the Biaffine parser [12]. Graphs are labeled as positive or negative sentiment.
- **Synthetic graph motif detection.** `BA2Motifs` [25] contains graphs with a Barabasi-Albert (BA) base graph of size 20 and a 5-node motif in each graph. Node features are 10-dimensional all-one vectors. The motif can be either a house-like structure or a cycle. Graphs are labelled in two classes based on which motif they contain.

**GNNs and explanation baselines.** We evaluate GStarX by explaining GCNs [19] on all datasets in our major experiment in Section 5.2. In the ablation study in Section 5.3, we further evaluate on GIN [40] and GAT [36] on certain datasets following [44]. All models are trained to convergence with hyperparameters and performance shown in Appendix A.2. We compare with 5 strong baselines representing the SOTA methods for GNN explanation: GNNExplainer [41], PGExplainer [25], SubgraphX [44], GraphSVX [9], and OrphicX [21]. In particular, SubgraphX and GraphSVX use Shapley-value-based scoring functions.

**Evaluation metrics.** Evaluating explanations is non-trivial due to the lack of ground truth. We follow [44, 43] to employ Fidelity, Inverse Fidelity (Inv-Fidelity), and Sparsity as our evaluation metrics. Fidelity and Inv-Fidelity measure whether the prediction is faithfully important to the model prediction by removing the selected nodes or only keeping the selected nodes respectively. Sparsity promotes fair comparison by controlling explanations to have similar sizes, since including more nodes generally improves Fidelity and Inv-Fidelity, and explanations with different sizes are not directly comparable. Ideal explanations should have high Fidelity, low Inv-Fidelity, and high Sparsity, indicating relevance and conciseness. Equations 9-11 show their formulas.

$$\text{Fidelity}(\mathcal{G}, g) = [f(\mathcal{G})]_{c^*} - [f(\mathcal{G}\backslash g)]_{c^*} \qquad (9)$$

$$\text{Inv-Fidelity}(\mathcal{G}, g) = [f(\mathcal{G})]_{c^*} - [f(g)]_{c^*} \qquad (10)$$

$$\text{Sparsity}(\mathcal{G}, g) = 1 - |g|/|\mathcal{G}| \qquad (11)$$

Fidelity and Inv-Fidelity are complementary and are both important for a good explanation $g$. Fidelity justifies the necessity for $g$ to be included to predict correctly. Inv-Fidelity justifies the sufficiency of a standalone $g$ to predict correctly. As they are analogous to precision and recall, we draw an analogy to the F1 score to propose a single-scalar-metric "harmonic fidelity" (H-Fidelity), where we normalize them by Sparsity and take their harmonic mean; see Appendix A.3 for the formula.

**Hyperparameters.** GStarX includes three hyperparameters: $\tau$ for the allocated surplus in the associated game, $m$ as the maximum graph size to perform exact HN value calculation, and $J$ as the number of samples for the MC approximation. In our experiments, we choose $\tau = 0.01$ since we need $\tau < \frac{2}{n}$ for convergence (Appendix E.2) and all graphs in the datasets above have less than 200 nodes. For $m$ and $J$, bigger values should be better for the MC approximation, and we found $m = 10$ and $J = n$ work well empirically.

## 5.2 Evaluation results

**Quantitative studies.** We report averaged test set H-Fidelity in Table 1. We conduct 8 different runs to get results with Sparsity ranging from 0.5-0.85 in 0.05 increments (Sparsity cannot be precisely guaranteed, hence it has minor variations across methods) and report the best H-Fidelity for each method. GStarX outperforms others on 4/6 datasets and has the highest average. We also follow [44] to show the Fidelity vs. Sparsity plots for all 8 sparsity in Appendix A.4.

**Qualitative studies.** We visualize the explanations of graphs in `GraphSST2` in Figure 2 and compare them qualitatively. We show explanations selected with high and comparable Sparsity on a positive (upper) graph and a negative (lower) graph. GStarX concisely captures the important words for sentiment classification without including extraneous ones for both sentences. Baseline methods generally select some-but-not-all important sentiment words, with extra neutral words as well. Among baselines, SubgraphX gives more reasonable results. However, it cannot cover two groups of important nodes with a limited budget because it can only select a connected subgraph as the explanation; e.g.

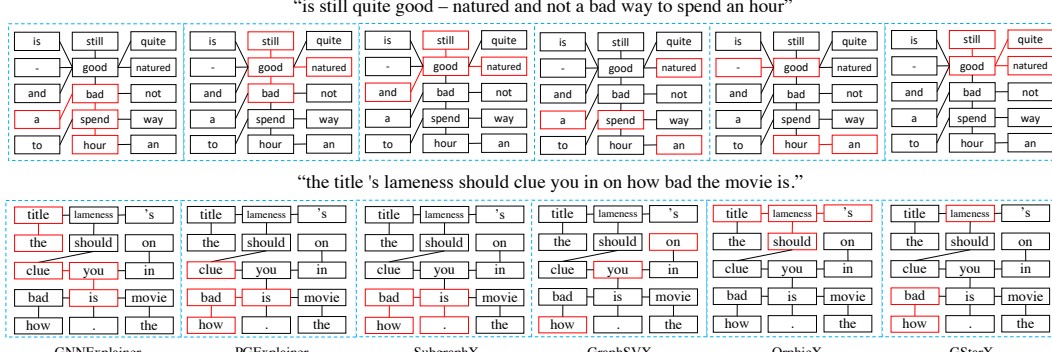

Figure 2: Explanations on sentences from `GraphSST2`. We show the explanation of one positive sentence (upper) and one negative sentence (lower). Red outlines indicate the selected nodes/edges as the explanation. GStarX identifies the sentiment words more accurately compared to baselines.

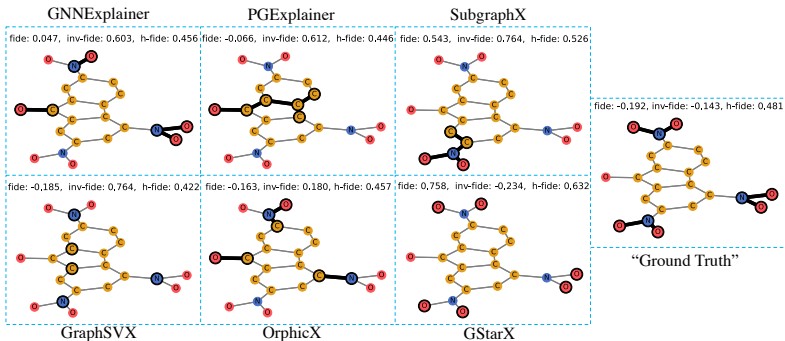

Figure 3: Explanations on a mutagenic molecule in `MUTAG`. Carbon atoms (C) are in yellow, nitrogen atoms (N) are in blue, and oxygen atoms are in red (O). Dark outlines indicate the selected nodes/edges as the explanation. We report the explanation Fidelity (fide), Inv-Fidelity (inv-fide), and H-Fidelity (h-fide). GStarX gives a significantly better explanation than other methods in terms of these metrics.

to cover the negative word "lameness" in the lower sentence, SubgraphX needs at least three more nodes along the way, which will significantly decrease Sparsity while including undesirable, neutral words. Moreover, we discussed in Section 3.2 that the Shapley value will downgrade the positive importance of the word "good" for the upper sentence. Comparing the normalized contribution scores of our HN-value-based method GStarX and the Shapley-based method GraphSVX, contribution of "good" is higher in ours: 0.1152 vs. 0.0371.

We visualize explanations selected with high and comparable Sparsity of a mutagenic molecule from `MUTAG` in Figure 3. Explanations on chemical graphs are harder to evaluate than text graphs as they require domain knowledge. `MUTAG` has been widely used as a benchmark for evaluating GNN explanations because human experts recognize -NO2 as mutagenic [8], which makes `MUTAG` a dataset with "ground truth"[3]. Surprisingly, we found that GStarX generates much better H-Fidelity/Fidelity/Inv-Fidelity than other methods and even the "ground truth" by only selecting the -O in -NO2 as explanations. In particular, the -0.234 Inv-Fidelity of GStarX means the selected subgraph has an even better prediction result than the original whole graph (0 Inv-Fidelity) and the ground truth (-0.143 Inv-Fidelity) because nodes not significant to the GNN prediction are removed. Fidelity metrics of baselines are inferior to GStarX because they include other non-discriminative carbon atoms despite they capture -NO2 to some extent. This suggests that even though **human experts identify -NO2** as the "ground truth" of mutagenicity, the **GNN only needs -O** to classify mutagenic molecules. With the goal being understand model behavior, GStarX explanation is better. Moreover, SubgraphX is the only baseline that has better H-Fidelity than the "ground truth", but it

---

[3]Carbon rings were also claimed as mutagenic by human experts, but we found it is not discriminative as they exist in both mutagenic and non-mutagenic molecules in `MUTAG`.

Table 1: The best H-Fidelity (higher is better) of 8 different Sparsity for each dataset. GStarX shows higher H-Fidelity on average and on 4/6 datasets.

| Dataset | GNNExplainer | PGExplainer | SubgraphX | GraphSVX | OrphicX | GStarX |
|---------|-------------|-------------|-----------|----------|---------|--------|
| BA2Motifs | 0.4841 | 0.4879 | **0.6050** | 0.5017 | 0.5087 | 0.5824 |
| BACE | 0.5016 | 0.5127 | 0.5519 | 0.5067 | 0.4960 | **0.5934** |
| BBBP | 0.4735 | 0.4750 | **0.5610** | 0.5345 | 0.4893 | 0.5227 |
| GraphSST2 | 0.4845 | 0.5196 | 0.5487 | 0.5053 | 0.4924 | **0.5519** |
| MUTAG | 0.4745 | 0.4714 | 0.5253 | 0.5211 | 0.4925 | **0.6171** |
| Twitter | 0.4838 | 0.4938 | 0.5494 | 0.4989 | 0.4944 | **0.5716** |
| Average | 0.4837 | 0.4934 | 0.5569 | 0.5114 | 0.4952 | **0.5732** |

Table 2: GStarX shows higher H-Fidelity for both GAT on GraphSST2 and GIN on MUTAG.

| Dataset | GNNExplainer | PGExplainer | SubgraphX | GraphSVX | OrphicX | GStarX |
|---------|-------------|-------------|-----------|----------|---------|--------|
| GraphSST2 | 0.4951 | 0.4918 | 0.5484 | 0.5132 | 0.4997 | **0.5542** |
| MUTAG | 0.5042 | 0.4993 | 0.5264 | 0.5592 | 0.5152 | **0.6064** |

can only capture one -NO2 because its search algorithm requires the explanation to be connected, so its Inv-Fidelity is not optimal. In fact, GNNExplainer, PGExplainer, and SubgraphX can never generate explanations including only disconnected -O without -N like GStarX, because the former two solve the explanation problem by optimizing edges (as opposed to Equation 4), and the latter requires connectedness. More MUTAG explanation visualizations are in Appendix H.

### 5.3 Ablation study and analysis

**Model-agnostic explanation.** GStarX makes no assumptions about the model architecture and can be applied to explain various GNN backbones. We use GCN for all datasets in the major experiment above for consistency, and we now further investigate performance on two more popular GNNs: GIN and GAT. We follow [44] to train GIN on MUTAG and GAT on GraphSST2[4], and show results in Table 2. For both settings, GStarX outperforms the baselines, which is consistent with results on GCN.

**Efficiency study.** The GStarX algorithm scales in $O(J)$ with practical $J \propto |\mathcal{V}|$. Following [44], we study the empirical efficiency of GStarX by explaining 50 randomly selected graphs from BBBP. We report the average run time in Table 3. Our results for the baselines are similar to [44]. GStarX is not the fastest method, but it is more than two times faster than SubgraphX. Since explanation usually doesn't have strict efficiency requirements in real applications, considering GStarX generates higher-quality explanations than the baselines, we believe the time complexity of GStarX is acceptable.

**Explanation sparsity study.** To further study whether the obtained scores by GStarX are sparse, we follow [11] to evaluate an entropy-based sparsity measure on model output scores. We show the average GStarX entropy-based sparsity on all datasets, and compare them with three reference score distributions on all $n$ nodes in a graph. 1) An upper bound: Uniform(n), which represents the least sparse output. 2) A practical lower bound: Uniform(0.25*n) which represents very sparse outputs with only top 25% of nodes. 3) Poisson(0.25*n), which is a more realistic version of case 2). Results in Table 4 show the average entropy-based sparsity of GStarX is much lower than Uniform(n) and close to Poisson(0.25*n), which justifies the GStarX outputs are indeed sparse. A more detailed discussion of this metric and these three reference distributions is in Appendix A.5.

## 6 Related work

**GNN explanation** aims to produce an explanation for a GNN prediction on a given graph, usually as a subgraph induced by important nodes or edges. Many existing methods work by scoring nodes or edges and are thus similar to this work. For example, the scoring function of GNNExplainer [41] is the mutual information between a masked graph and the prediction on the original graph, where soft masks on edges and node features are generated by direct parameter learning. PGExplainer [25] uses the same scoring function as [41] but generates a discrete mask on edges by training an edge mask predictor. SubgraphX [44] uses the Shapley value as its scoring function on subgraphs

---

[4]As some baselines take over 24 hours on full GraphSST2, we randomly select 30 graphs for this analysis.

Table 3: Average running time on 50 graphs in BBBP

| Method | GNNExplainer | PGExplainer | SubgraphX | GraphSVX | OrphicX | GStarX |
|--------|--------------|-------------|-----------|----------|---------|--------|
| Time(s) | 11.92 | 0.03 (train 720) | 75.96 | 3.06 | 0.15 (train 915) | 31.24 |

Table 4: The entropy-based sparsity scores of GStarX vs. three reference distributions, which shows GStarX outputs are indeed sparse.

| Dataset | BA2Motifs | BACE | BBBP | GraphSST2 | MUTAG | Twitter |
|---------|-----------|------|------|-----------|-------|---------|
| GStarX | 2.1352 | 2.4481 | 2.3290 | 2.3282 | 2.2434 | 2.2114 |
| Uniform(n) | 3.2189 | 3.5080 | 3.0728 | 2.8698 | 2.8612 | 2.9833 |
| Uniform(0.25*n) | 1.8326 | 2.1217 | 1.6893 | 1.4855 | 1.4749 | 1.5970 |
| Poisson(0.25*n) | 2.3204 | 2.4686 | 2.2416 | 2.1336 | 2.1323 | 2.1945 |

selected by Monte Carlo Tree Search (MCTS), and GraphSVX [9] uses a least-square approximation to the Shapley value to score nodes and their features. While SubgraphX and GraphSVX were shown to perform better than prior alternatives, as we show in Section 3, the Shapley value they try to approximate is non-ideal as it is non-structure-aware. Although SubgraphX and GraphSVX use $L$-hop subgraphs and thus technically they use the graph structure, such structure usage are very limited in achieving structure-awareness as we show in Appendix G. While there are many other GNN explanation methods from very different perspectives, i.e. gradient analysis [28], model decomposition [1], surrogate models [37], and causality [20, 21], we defer their details to Appendix C given their lesser relevance.

**Cooperative game theory** originally studies how to allocate payoffs among a set of players in a cooperative game. Recently, certain ideas from this domain have been successfully used in feature importance scoring for ML model explanation [22, 32, 24]. When used for model explanation, data features becomes players in the game, e.g. pixels for images, and the value of the game gives feature importance scores. The vast majority of works in this line, like the ones cited above, deem the Shapley value [30] to be the only choice. In fact, there are many other values with different properties and used in different situations in cooperative game theory. However, to the best of our knowledge, only [4] mentions the Myerson value [26] in the context of proposing a connected Shapley (C-Shapley) value for explaining sequence data, and it is not directly comparable to ours for graph data. A detailed discussion of the Myerson value and the C-Shapley value can be found in Appendix F. Our work follows the cooperative game theory approach to explain models on graph data using the HN value [15], which as we show is a better choice than the Shapley value given its structure-awareness.

## 7   Conclusion and future work

In summary, we study GNN explanation on graphs via node importance scoring. We identify the non-structure-aware challenge of existing Shapley-value-based approaches and propose GStarX to assign importance scores to each node via a structure-aware HN value. We also build connections between the HN value surplus allocation and GNN message passing. GStarX demonstrates its superiority over strong baselines on chemical and text graph classifications. A limitation of GStarX is that the importance of different node feature dimensions is not explained. One future work is to add this extension, which could be done by scoring a subset of nodes together with a subset of features each time. Another future direction is to exploit the rich cooperative game theory literature. Beyond the Shapley value, more values are possible for explaining ML models. For graph data, edge-based values like [2] can potentially be applied to an alternative edge-based objective like Equation 4. Other values may be appropriate to more data types beyond graphs.

## Acknowledgement

This work was partially supported by NSF III-1705169, NSF 1937599, NSF 2119643, Okawa Foundation Grant, Amazon Research Awards, Cisco research grant USA000EP280889, Picsart Gifts, and Snapchat Gifts.

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
