# A    Experiment details

## A.1    Dataset statistics

In Table 5, we provided the statistics of all datasets used in our experiments.

Table 5: Dataset Statistics.

| Dataset | # Graphs | # Test Graphs | # Nodes (avg) | # Edges (avg) | # Features | # Classes |
|---------|----------|---------------|---------------|---------------|------------|-----------|
| MUTAG | 188 | 20 | 17.93 | 19.79 | 7 | 2 |
| BACE | 1,513 | 152 | 34.01 | 73.72 | 9 | 2 |
| BBBP | 2,039 | 200 | 24.06 | 25.95 | 9 | 2 |
| GraphSST2 | 70,042 | 1821 | 9.20 | 10.19 | 768 | 2 |
| Twitter | 6,940 | 692 | 21.10 | 40.20 | 768 | 3 |
| BA2Motifs | 1,000 | 100 | 25 | 25.48 | 10 | 2 |

## A.2    Model architectures and implementation

In Table 6, we provided the hyperparameters and test accuracy for the GCN model used in our major experiments. In Table 2, we provided the hyperparameters and test accuracy for the GIN and GAT model used in our analysis experiment. Most parameters are following [44], with small changes to further boost the test accuracy.

We run all experiments on a machine with 80 Intel(R) Xeon(R) E5-2698 v4 @ 2.20GHz CPUs, and a single NVIDIA V100 GPU with 16GB RAM. Our implementations are based on Python 3.8.10, PyTorch 1.10.0, PyTorch-Geometric 1.7.1 [10], and DIG [23]. We adapt the GNN implementation and most baseline explainer implementation from the DIG library, except for GraphSVX and OrphicX where we adapt the official implementation. For the baseline hyperparameters, we closely follow the setting in [44] and [9] for a fair comparison. Please refer to [44] Section 4.1 and [9] Appendix E for details.

Table 6: GCN architecture hyperparameters according to results in Table 6

| Dataset | #Layers | #Hidden | Pool | Test Acc |
|---------|---------|---------|------|----------|
| BA2Motifs | 3 | 20 | mean | 0.9800 |
| BACE | 3 | 128 | max | 0.8026 |
| BBBP | 3 | 128 | max | 0.8634 |
| MUTAG | 3 | 128 | mean | 0.8500 |
| GraphSST2 | 3 | 128 | max | 0.8808 |
| Twitter | 3 | 128 | max | 0.6908 |

Table 7: GIN and GAT architecture hyperparameters according to results in Table 2. For GAT, we use 10 attention heads with 10 dimension each, and thus 100 hidden dimensions.

| Dataset | #Layers | #Hidden | Pool | Test Acc |
|---------|---------|---------|------|----------|
| GraphSST2(GAT) | 3 | 10 ×10 | max | 0.8814 |
| MUTAG(GIN) | 3 | 128 | max | 1.0 |

## A.3    Exact formula for evaluation metrics

Formulas for Fidelity, Inv-Fidelity, and Sparsity are shown in Equation 9, 10, and 11. In Equation 12, 13, and 14, we show formulas for normalized fidelity (N-Fidelity), normalized inverse fidelity (N-Inv-Fidelity), and harmonic fidelity (H-Fidelity). Both the N-Fidelity and N-Inv-Fidelity are in $[-1, 1]$. The H-Fidelity flips N-Inv-Fidelity, rescales both values to be in $[0, 1]$, and takes their harmonic mean.

$$\text{N-Fidelity}(\mathcal{G}, g) = \text{Fidelity}(\mathcal{G}, g) \cdot (1 - \frac{|g|}{|\mathcal{G}|}) \tag{12}$$

$$\text{N-Inv-Fidelity}(\mathcal{G}, g) = \text{Inv-Fidelity}(\mathcal{G}, g) \cdot \left(\frac{|g|}{|\mathcal{G}|}\right) \tag{13}$$

Let $m1 = \text{N-Fidelity}(\mathcal{G}, g)$, $m2 = \text{N-Inv-Fidelity}(\mathcal{G}, g)$

$$\begin{aligned}
\text{H-Fidelity}(\mathcal{G}, g) &= \frac{2}{\left(\frac{1+m1}{2}\right)^{-1} + \left(\frac{1-m2}{2}\right)^{-1}} \\
&= \frac{(1 + m1) \cdot (1 - m2)}{(2 + m1 - m2)}
\end{aligned} \tag{14}$$

## A.4 Fidelity vs. sparsity plots

In Table 1, we report the best H-Fidelity among 8 different sparsities for each method on each dataset. We also follow [44] to show the Fidelity vs. Sparsity plots in Figure 4 row1. Note that GraphSVX tends to give sparse explanations on some datasets, we still pick 8 different sparsities for it but mostly on the higher end. We also show the *1 - Inv-Fidelity* vs. sparsity plots and the H-Fidelity vs. sparsity plots. Curves in all three plots are the higher the better.

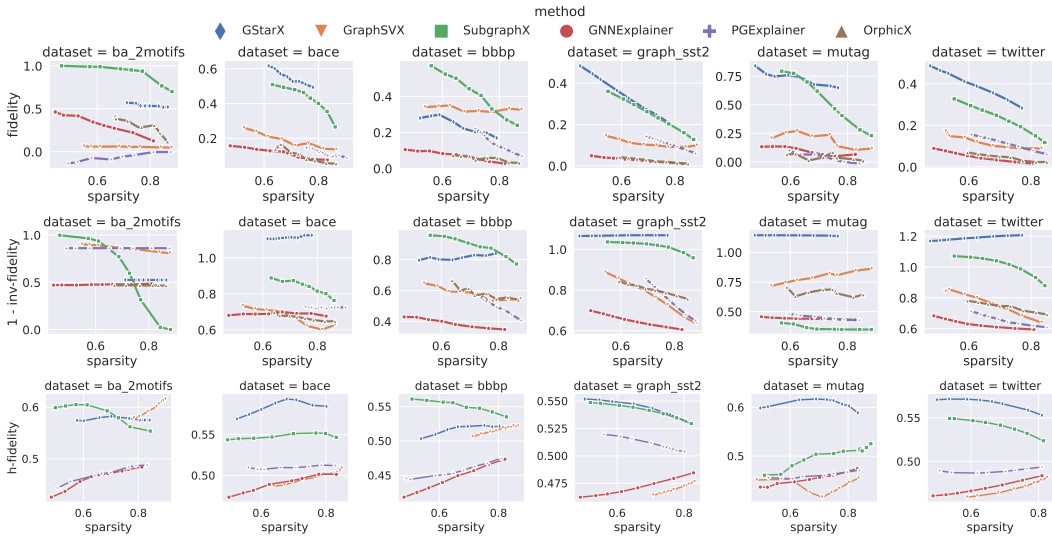

Figure 4: Fidelity (row1), 1 - Inv-Fidelity (row2), and H-Fidelity (row3) vs. Sparsity on all datasets corresponding to the results shown in Table 1. All three metrics are the higher the better. We see that GStarX outperforms the other methods

## A.5 Detailed entropy-based sparsity evaluation

In Section 5.2 we study whether the obtained scores by GStarX are sparse and follow [11] to apply an entropy-based sparsity measure on scores. We now provide a more detailed discussion of this study.

The entropy-based sparsity, as defined in Definition 2 in [11], is shown in the Equation 15 below. Here $\phi$ is the model output scores for a data instance, and $\tilde{\phi}_i = \frac{\phi_i}{\sum_i \phi_i}$ represent normalized scores.

$$H(\tilde{\phi}) = -\sum_{i \in n} \tilde{\phi}_i \log \tilde{\phi}_i \tag{15}$$

The entropy-based sparsity helps us to understand how sparse an explanation is, before the scores are turned into hard explanation by thresholding or selecting top k. In Table 4, we show the average scores for GStarX on all datasets, and compare them with three reference cases. 1) The entropy of uniform distribution over all n nodes in a graph, i.e., Uniform(n), which represents the least sparse output and is an upper bound of entropy-based sparsity. 2) The entropy of uniform distribution over the top 25% nodes in a graph, i.e., Uniform(0.25*n), where probabilities of the bottom 75% nodes

are set to zero. This case is very sparse since 75% of nodes are deterministically excluded, which can be treated as a practical lower bound of entropy-based sparsity. 3) The entropy of Poisson distribution with mean 0.25*n, i.e. Poisson(0.25*n), which is a more realistic version of the sparse output in case 2). Instead of setting all 75% of nodes to have probability zero, we assume the probabilities for tail nodes decrease exponentially as a Poisson distribution while the mean is kept the same as in case 2). Results in Table 4 show that the average entropy-based sparsity of GStarX is between Uniform(0.25*n) and Uniform(n) and close to Poisson(0.25*n), which justifies the GStarX outputs are indeed sparse.

## B    GStarX for node classification

Even though the GStarX algorithm is stated for graph classification, it works for node classification as well. This can be easily seen as the GNN node classification can be covert to classify an ego-graph. Given a graph $\mathcal{G}$ with $\mathcal{V} = \{u_1, \ldots, u_n\}$. Node classification on $u_i$ with an $L$-layer GNN can be converted to a graph classification. The target graph to classify will be the $L$-hop ego-graph centered at $u_i$, because this is the receptive field of the GNN for classifying $u_i$ and nodes further away won't influence the result. The label of the graph will be the label of $u_i$. In this case, the final readout layer of the GNN will be indexing $u_i$ instead of pooling. Given this kind of conversion, everything we showed in Section 4 follows.

## C    More related work

**GNN explanation continued**    Besides the perturbation-based method we mentioned in Section 6, there are several other types of approaches for GNN explanation. Gradient-based methods are widely used for explaining ML models on images and text. The key idea is to use the gradients as the approximations of input importance. Such methods as contrastive gradient-based (CG) saliency maps, Class Activation Mapping (CAM), and gradient-weighted CAM (Grad-CAM) have been generalized to graph data in [28]. Decomposition-based methods are a popular way to explain deep NNs for images. They measure the importance of input features by decomposing the model predictions and regard the decomposed terms as importance scores. Decomposition methods including Layer-wise Relevance Propagation (LRP) and Excitation Backpropagation (EB) have also been extended to graphs [28, 1]. Surrogate-based methods work by approximating a complex model using an explainable model locally. Possible options to approximate GNNs include linear model as in GraphLIME [17], additive feature attribution model with the Shapley value as in GraphSVX [9], and Bayesian networks as in [37]. GNN explainability has also been studied from the causal perspective. In [20, 21], generative models were constructed to learn causal factors, and explanations were produced by analyzing the cause-effect relationship in the causal graph.

## D    Properties of the Shapley value

The Shapley value was proposed as the unique solution of a game $(N, v)$ that satisfies three properties shown below, i.e. *efficiency*, *symmetry*, and *additivity* [30]. These three properties together are referred as an axiomatic characterization of the Shapley value. The *associated consistency* properties introduced in Section 4.1 provides a different axiomatic characterization.

**Property D.1** (**Efficiency**).
$$\sum_{i \in N} \phi_i(N, v) = v(N)$$

**Property D.2** (**Symmetry**). If $v(S \cup \{i\}) = v(S \cup \{j\})$ for all $S \in N \backslash \{i, j\}$, then
$$\phi_i(N, v) = \phi_j(N, v)$$

**Property D.3** (**Additivity**). Given two games $(N, v)$ and $(N, w)$,
$$\phi(N, v + w) = \phi(N, v) + \phi(N, w)$$

The efficiency property states that the value should fully distribute the payoff of the game. The symmetry property states that if two players make equal contributions to all possible coalitions formed by other players (including the empty coalition), then they should have the same value. The additivity property states that the value of two independent games should be added player by player. It is the most useful for a system of independent games.

# E  Properties and calculation of the HN value

## E.1  Consistency and associated games

One reason for the Shapley value's popularity is its *axiomatic characterization*, indicating that it is the unique solution that satisfies a set of desirable properties (see Appendix D). Then [14] proposed a new axiomatic characterization of the Shapley value based on a different *associated consistency* property. The *consistency* property is a common analysis tool used in game theory [16, 7, 31, 27]. The idea is to analyze a game $(N, v)$ by defining other reduced games $(S, v_S)$ for $S \subseteq N$, and a solution function $\phi$ is called *consistent* when $\phi(N, v)$ yields the same payoff as $\phi(S, v_S)$ on each $S$. When $(S, v_S)$ is defined with desired properties, these good properties can be enforced for a solution by requiring consistency. The associated consistency in [14] is a special case of consistency between $(N, v)$ and only one other game $(N, v^*)$, which is called the *associated game*. [14] shows that a carefully designed associated game uniquely characterizes the Shapley value. Associated consistency is also the key idea of the HN value.

## E.2  Limit game and the axiomatic characterization

The HN value is established on a special associated game as we discussed in Section 4.1. We can actually write this associated game in a more compact matrix form, where we slightly abuse notation and use $v$ and $v_\tau^*$ to represent vectors of payoffs for all $S \subseteq N$ under the original and associated game respectively. In other words, $v(S)$, which is used to represent evaluating the coalition $S$ using the characteristic function $v$, now can also be interpreted as indexing the vector $v$ with index $S$.

**Lemma E.1.** *A matrix form of the associated game $(N, v_\tau^*, \mathcal{G})$ is given by*

$$v_\tau^* = \boldsymbol{H}_{\{\tau, n, \mathcal{G}\}} v \tag{16}$$

The matrix $\boldsymbol{H}_{\{\tau, n, \mathcal{G}\}}$ depends on the hyperparameter $\tau$, number of players $n$, and the graph $\mathcal{G}$. When these variables are clear from the context, we drop them and write $v_\tau^* = \boldsymbol{H} v$. Please refer to [15] for the proof of Lemma E.1.

With the matrix form, we can define the limit game.

**Definition E.2.** *Given a game $(N, v, \mathcal{G})$, its limit game $(N, \tilde{v}, \mathcal{G})$ is defined by*

$$\tilde{v} = \lim_{p \to \infty} \boldsymbol{H}^p v \tag{17}$$

Notice that although the matrix $\boldsymbol{H}$ is constructed from the associated game and depends on $\tau$, the powers of $\boldsymbol{H}$ actually converge to a limit independent from $\tau$, when $\tau$ is sufficiently small. The general condition depends on the actual graph, but $0 < \tau < \frac{2}{n}$ is proven to be sufficient for the complete graph case [14]. As we discussed in Section 4.1, the limit game can be seen as constructing associated games repeatedly until the characteristic function converges.

An axiomatic characterization of the HN value regarding its uniqueness is given by the following theorem based on the limit game. The associated consistency is the core property related to this work. We encourage the readers to check [15] for the other two properties.

**Theorem E.3.** *There exists a unique solution $\phi$ that verifies the associated consistency, i.e. $\phi_i(N, v, \mathcal{G}) = \phi_i(N, v_\tau^*, \mathcal{G})$, inessential game, and continuity. $\phi$ is given by*

$$\phi_i(N, v, \mathcal{G}) = \tilde{v}(\{i\}) \tag{18}$$

## E.3  The algorithm for computing the HN value

We show the algorithm for Compute-HN-MC (Algorithm 3) mentioned in Section 4.3. The algorithm is a combination of Equation 16, 17, and 8.

# F  The Myerson value and the C-Shapley value

## F.1  The Myerson value

In the study of cooperative games, [26] proposed to characterize the cooperation possibilities between players using a graph structure $\mathcal{G}$, which leads to the communication structure introduced in Section

**Algorithm 3** The Compute-HN-MC Function

---

**Input:** Graph instance $\mathcal{G}$ with nodes $\mathcal{V} = \{u_1, \ldots, u_n\}$, characteristic function $v$, hyperparameter $\tau$, maximum sample size $m$, number of samples $J$
Let $\psi_1, \ldots, \psi_n$ be $n$ empty lists
**for** $j = 1$ **to** $J$ **do**
    Sample $g_{S^j}$ from $\mathcal{G}$ s.t. $S^j = \{u_{j_1}, \ldots, u_{j_l}\}$ and $l < m$
    $\phi^j = $ Compute-HN$(g_{S^j}, S^j, v(\cdot), \tau)$
    **for** $k = 1$ **to** $l$ **do**
        Append $\phi_k^j$ to $\psi_{j_k}$
    **end for**
**end for**
Set $\phi_i$ to be the mean of $\psi_i$
**Return:** $\phi$

---

2.2 and the Myerson value as a solution for this special type of games $(N, v, \mathcal{G})$. The Myerson value is closely related to the Shapley value. In fact, it is the Shapley value on a transformed game where players are partitioned by the graph. We now formally introduce the partition and the transformed game.

**Definition F.1 (Partition).** Given a set of players $N$ and a graph $\mathcal{G}$. For any coalition $S \subseteq N$, the partition of $S$ is denoted by $S/\mathcal{G}$ and defined by

$$S/\mathcal{G} = \{\{i | i \text{ and } j \text{ are connected in S by } \mathcal{G}\} | j \in S\}$$

and a member of the set $S/\mathcal{G}$ is called a component of $S$.

**Definition F.2 (Transformed Game).** Given a game $(N, v, \mathcal{G})$, we can transform it to a new game $v/\mathcal{G}$ such that for all $S \subseteq N$

$$(v/\mathcal{G})(S) = \sum_{T \in S/\mathcal{G}} v(T)$$

Intuitively, given a coalition $S$, the transformed game treats each connected component of $S$ as independent, evaluates them separately, and sums their payoff as the payoff of $S$.

The Shapley value has an axiomatic characterization that uniquely determines it as we introduced in Appendix D. Likewise, the Myerson value was proposed to be a unique solution that satisfies the *component efficiency* and the *fairness* property defined below.

**Property F.3 (Component Efficiency).** For a game $(N, v, \mathcal{G})$ and any connected component $S \in N/\mathcal{G}$, a solution is component efficient if

$$\sum_{i \in S} \phi_i(N, v, \mathcal{G}) = v(S)$$

**Property F.4 (Fairness).** For a game $(N, v, \mathcal{G})$ and any edge $(i, j)$ in $\mathcal{G}$, let $\tilde{\mathcal{G}}$ be $\mathcal{G}$ with the edge $(i, j)$ removed, a solution is fair if

$$\phi_i(N, v, \mathcal{G}) - \phi_i(N, v, \tilde{\mathcal{G}}) = \phi_j(N, v, \mathcal{G}) - \phi_j(N, v, \tilde{\mathcal{G}})$$

The component efficiency property is an extension of the regular efficiency property to games with a communication structure. It requires efficiency to hold for each disconnected piece because these pieces are assumed as independent from each other. The fairness property states that if breaking an edge $(i, j)$ changes the value of player $i$, then the value of player $j$ should be changed by the same amount.

**Theorem F.5 (Myerson Value).** *There exists a unique solution $\phi$ of game $(N, v, \mathcal{G})$ satisfying component efficiency and fairness. With $\tilde{\phi}$ represents the Shapley value, the solution is given by the formula*

$$\phi(N, v, \mathcal{G}) = \tilde{\phi}(N, (v/\mathcal{G}))$$

For games with a communication structure, the Myerson value is a better choice than the Shapley value as it uses the graph structure. However, it also suffers from some criticisms. For example, the

fairness assumption may not be realistic. When an existing edge is broken, the value changes for players on the two edge ends can be asymmetric. Intuitively, if the edge connects a popular hub player $i$ to a leaf player $j$, then the change of $i$ can be less significant than $j$ since $j$ becomes isolated when $(i, j)$ is removed. This is also the case when the game value is used for model explanation. For example in Figure 1 (b), when the edge ("good", "quite") is broken, the value of "quite" should change a lot. It used to contribute positively together with "good", and thus gets some payoff allocation, but it now becomes an isolated node, which is neutral by itself. On the other hand, the word "good" can still contribute positively by itself and interact with other nodes through its other edges, and thus its value shouldn't change too much. Because of such criticisms, we choose to use the HN value as our scoring function, which characterizes the value by associated consistency rather than fairness.

## F.2 The C-Shapley value

The Myerson value was also mentioned in [4] for the model explanation on text, where the C-Shapley value was proposed as an approximation of the Shapley value, and it was claimed to be equal to the Myerson value. We have discussed why Shapley value and Myerson are not-ideal choices for explaining graph data in Section 3 and Appendix F.1. These are partially the reason why our HN-value-based method is better than the C-Shapley value. However, the major reason why we don't do a direct comparison to the C-Shapley value as a baseline is that its formula only works for line graphs like sequence data, and not even all nodes in line graphs. In contrast, our target task is general graph prediction for graphs with possibly complicated topological structures.

We now clarify a mistake of the C-Shapley value formula and explain why it won't work for general graphs. The notations are following the [4], where $d$ is the number of players corresponding to $n$ in our notation, and $[d]$ corresponding to $N$.

The formula for the C-Shapley value is given in Equation 6 in Definition 2 in the paper, and it is stated for "a graph G" without mentioning any assumptions of the graph. However, from the proof of this formula in Appendix B.2 in the paper, the line graph assumption can be seen in two places. The first place is Equation 20, where the set $\mathcal{C}$ is explicitly defined only for subsequences. The second place is Equation 22, the first line converts $\sum_{A:U_S(A)=U}$ to $\sum_{i=0}^{d-|U|-2}$, which is implicitly saying $V_S(A)$ can be picked from all $d$ but $|U| - 2$ nodes. However, this conversion is only possible when there are exactly 2 edges between $U$ and $[d] \backslash U$, i.e. the middle part of a line graph. If there are $l$ edges between $U$ and $[d] \backslash U$, then the summation should go up to $d - |U| - l$. When $l = 0$, i.e. $U$ equals $[d]$ or a connected component of $[d]$, no partition is needed and the coefficient simply evaluates to 1. By correcting all these cases, the final formula for the C-Shapley value coefficients of marginal contributions thus becomes

$$\sum_{i=0}^{d-|U|-l} \frac{1}{\binom{d-1}{i+|U|-1}} \binom{d-|U|-l}{i} \tag{19}$$

$$= \frac{d}{(|U|+l)\binom{|U|+l-1}{|U|-1}} \tag{20}$$

$$= \frac{dl}{(|U|+l)(|U|+l-1)\cdots|U|} \tag{21}$$

for $l > 0$, and 1 for $l = 0$.

The correct formula for the C-Shapley value of general graphs will be

$$\phi_X(i) = \begin{cases} \sum_{U \in \mathcal{C}} \frac{l}{(|U|+l)\cdots|U|} m_X(U, i) & \text{if } l > 0 \\ \frac{1}{d} & \text{if } l = 0 \end{cases} \tag{22}$$

with $l$ represents the edges between $U$ and $[d] \backslash U$ and $\mathcal{C}$ represents all connected subgraphs in $[d]$ containing $i$.

To verify this formula with the 3-node toy graph in Figure 5. When computing the value of node 0 (left), the three connected components containing 0 are $\mathcal{C} = \{\{0\}, \{0, 1\}, \{0, 1, 2\}\}$. Since 0 is an

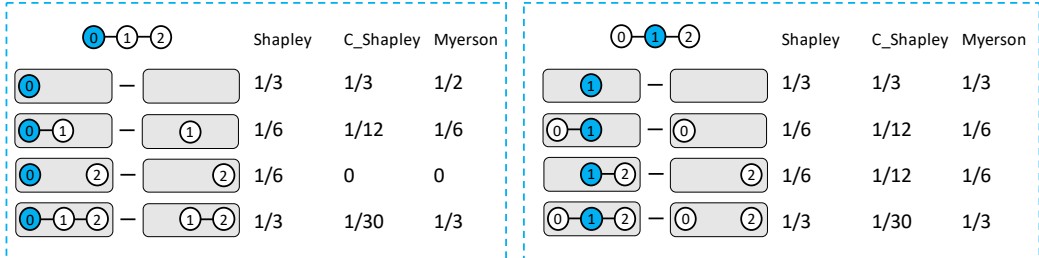

Figure 5: A toy 3-node graph example for comparing the mariginal contribution coefficients between the Shapley, the C-Shapley, and the Myerson value. **(a)** Value computation for node 0 (left). **(b)** Value computation for node 1 (right).

end node and has no leaf nodes to its left, $l$ for these three components will be 1, 1, and 0 respectively. According to our new formula in Equation 22, the coefficients will be $\frac{1}{2}$, $\frac{1}{6}$, and $\frac{1}{3}$ respectively, with the disconnected $\{0, 2\}$ case removed. This matches the original idea of Myerson value, where the $\{0, 2\} - \{2\}$ case is reduced to the $\{0\} - \emptyset$ case, which turns the Shapley coefficients from $[\frac{1}{3}, \frac{1}{6}, \frac{1}{6}, \frac{1}{3}]$ to $[\frac{1}{3} + \frac{1}{6}, \frac{1}{6}, \frac{1}{6} - \frac{1}{6}, \frac{1}{3}]$, which is $[\frac{1}{2}, \frac{1}{6}, 0, \frac{1}{3}]$. However, the original C-Shapley formula from Equation 6 in the [4] evaluates to $[\frac{1}{3}, \frac{1}{12}, 0, \frac{1}{30}]$, which doesn't match the Myerson value and not even sum up to 1. Another example of computing the value of node 1 is shown in Figure 5 right.

The C-Shapley, even with the correct formula, eventually boils down to an approximation of the Shapley value or the Myerson value, which as we discussed are less ideal than the HN value. Also, the correct formula in Equation 22 requires generating all possible subgraphs $U$ containing the node $i$ and specify the edges between $U$ and $[d]\backslash U$. This makes the computation very complicated, we thus skip the comparison to the C-Shapley value.

## G   Use the graph structure via an L-hop cutoff

Although the Shapley value itself is not structure-aware, we do note the existing Shapley-value-based GNN explanation methods use an L-hop cutoff to help approximate the Shapley value [44, 9]. Technically, this operation uses the graph structure, so we can't strictly refer to these explanation methods as not structure-aware. However, we argue that the L-hop cutoff is a naive way of utilizing the graph structure. It has several concerns, and it is not the same structure-aware as the HN value.

The L-hop cutoff approximates the Shapley value of node $i$ by considering only the L-hop neighbors of $i$ when explaining an L-layer GNN. The rationale of this operation is that an L-layer GNNs only propagate messages within L-hops so a node more than L-hop away from $i$ has never passed any messages to $i$ which means no interactions are possible. In existing Shapley-value-based GNN explanation methods, this L-hop cutoff operation was meant for reducing the exponentially growing computations of the Shapley value, and the ultimate goal is still to compute the Shapley value. The L-hop cutoff operation has several issues making it a less desirable choice. **1)** Even meant to save computation, there are still many nodes involved in the computation after applying the L-hop cutoff since the number of nodes grows exponentially as L grows. For advanced GNNs, the L can be large. When L is larger than the diameter of the graph, which is actually the case for many recent deep GNNs, the L-hop cutoff is not effective anymore. **2)** When constructing coalitions of nodes within the local graph of L-hops, the computation still follows the Shapley value formula. This means the useful graph structure information among these nodes is forfeited which causes the structure-awareness concern of Shapley value as we discussed in Section 3,

## H   More explanation visualizations

Under the same setting as Figure 3, we visualize more explanations in 6.

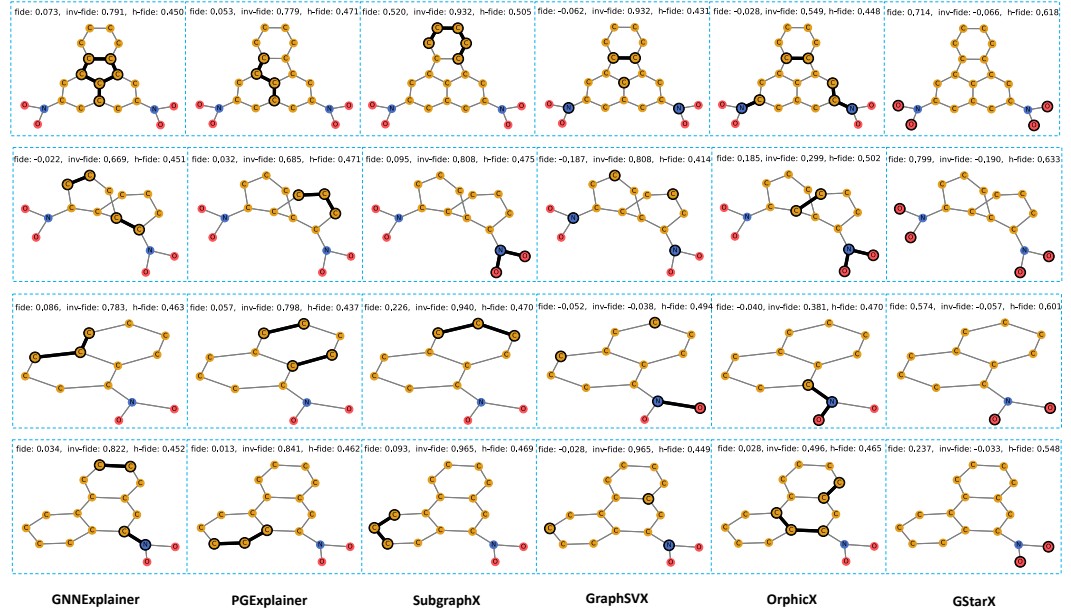

Figure 6: Explanations on a mutagenic molecule from the MUTAG dataset. Carbon atoms (C) are in yellow, nitrogen atoms (N) are in blue, and oxygen atoms (O) are in red. We use dark outlines to indicate the selected subgraph explanation and report the Fidelity (fide), Inv-Fidelity (inv-fide), and H-Fidelity (h-fide) of each explanation. GStarX gives a significant better explanation than other methods in terms of these metrics.

"occasionally funny , always very colorful and enjoyably overblown in the traditional almodóvar style."

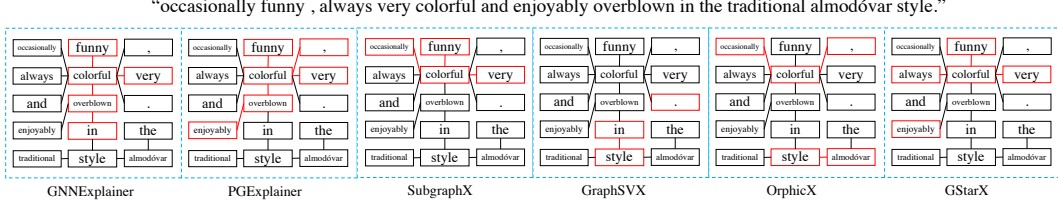

Figure 7: Explanations on sentences from GraphSST2. The sentence is predicted to be positive sentiment. Red outlines indicate the selected nodes/edges as the explanation. GStarX identifies the sentiment words more accurately compared to baselines.