# OpenReview forum: "GStarX: Explaining Graph Neural Networks with Structure-Aware Cooperative Games"
_NeurIPS.cc/2022/Conference — NeurIPS 2022 Accept_

### Official Review · Reviewer_ketX · 2022-07-07

**Rating:** 5
**Confidence:** 4
**Soundness:** 3 good
**Presentation:** 3 good
**Contribution:** 2 fair

**Summary:**

The explainability of GNNs is crucial for modeling graph data. This work is based on the HN value instead of the Shapley value to explain the GNNs, which involve the graph structure information.

**Questions:**

Please describe the original theoretical contribution of the paper.



**Strengths And Weaknesses:**

Pros:

The proposed method is quite clearly defined. Most of the time, I enjoyed reading this paper because of its clear and concise introduction to the preliminaries of this work (including Shapley value and HN value). Considering graph structure information via HN value to explain the GNNs is reasonable. The experiments have demonstrated the effectiveness of the proposed GStarX algorithm.

Cons:

One limitation of the approach is its lack of technical originality. Although using the HN value is interesting and innovative on this issue, this paper did not provide any original theoretical analysis on the benefit of leveraging graph structure information via HN value to explain the GNNs. Meanwhile, for the examples in Figure 1, only comparing the basic Shapley method is not enough to demonstrate the advantages over other SOTA methods. And, for Figure 1 (b), it is hard to say whether the coalition {“not”, “good”} contributes negatively when using the GNNs to model two unconnected nodes: “good” and “not”.

---

> ### Author Response · Authors · 2022-08-02
> **Response to Reviewer ketX**
>
> We truly appreciate your comments. Your suggestions make our paper clearer and stronger. We hope our response can address your questions and further clarify our contribution. Please find our detailed response below. The PDF is also revised accordingly with changes highlighted in red.
>
> > Please describe the original theoretical contribution of the paper.
>
> Indeed, our work relies on existing theoretical analysis from [1]; we would not claim our contribution to be theoretical in nature. Our major contribution is to propose an effective GNN explainer from the empirical perspective. Besides the explainer, we further connect strong theoretical tools from game theory with practical applications of GNNs. From this perspective, what we stated in Section 4.2, the connection between the HN value surplus allocation and the GNN message passing is our original contribution. Also, there is rich literature on many more values other than the Shapley value from cooperative game theory. Our work also serves the purpose of highlighting this avenue for interplay between the fields, and shifting focus beyond the original Shapley value, which has caught on due to its intuitiveness and existing introduction in machine learning literature; in reality, multiple such values can be explored depending on unique considerations of applications, for graphs or even more general data types. Given this enlightenment, we believe more theoretical studies are soon to come.
>
> > Examples in Figure 1 only compare to the basic Shapley method but not other SOTA methods. In Figure 1(b), the contribution of coalition {“not”, “good”} is not clear.
>
> We agree that by only looking at examples in Figure 1, it is not super convincing our method has a significant advantage over other SOTA methods. Our purpose in mentioning these examples is to illustrate the idea of “structure-awareness” and the difference between the Shapley value and the HN value. We assume most of the readers of our paper will have some familiarity with the Shapley value, but few of them will be familiar with the HN value. The advantage of our methods over the Shapley-value-based methods and other SOTA methods has been demonstrated both qualitatively and quantitatively in Section 5.2. In that section, we use visualization of important nodes, and metrics like fidelity/inverse fidelity to truly show the advantage of our method. Following your point, we have also added a clarification in Section 3.2 in the discussion of Figure 1, which points readers to Section 5.2 to observe the qualitative/quantitative impacts of this structure-awareness empirically. We appreciate your suggestion for helping us better illustrate this point.
>
> For the text graph in Figure 1(b), we have added a detailed study of it in Section 5.2 as well (updated Figure 2 and discussion in line 268 -271). In particular, we used inverse fidelity for answering your questions about the contribution of {“not”, “good”}. As defined in Equation (10) and stated below, inverse fidelity is the predicted probability of the whole graph minus the predicted probability of the selected-node-induced subgraph, which we believe is the clearest metric for answering your question.
>
> $invfidelity(G, g) = \left[ f(G) \right] _{c^*} - \left[ f(g) \right] _{c^*}$
>
> For the text graph in Figure 1(b), the inv-fidelity(G, {“good”}) = **– 0.023**. This negative value means the single node “good” actually generates a higher probability of being positive sentiment than the whole graph. The inv-fidelity(G, {“not”, “good”}) = **0.18**, which means the subgraph with “not” and “good” generates a lower probability of being positive sentiment than the whole graph. As you pointed out, the contribution of {“not”, “good”} is not too negative when they are disconnected, but its prediction is lower than {“good”} and G.
>
> Our added discussions in Section 5.2 further illustrate this point that the Shapley value will downgrade the positive contribution of the word “good” to the sentence. Comparing the normalized contribution score of our HN-value-based method GStarX and the Shapley-based method GraphSVX, the contribution of “good” is higher in ours: 0.1152 vs. 0.0371. In this case, GStarX gives the explanation {“good”, “natured”, “quite”, “still”}, which we believe is more reasonable than the explanation {“natured”, “a”, “spend”, “an”} produced by GraphSVX. Please find these explanation visualization in the updated Figure 2.
>
> [1] Gérard Hamiache and Florian Navarro. Associated consistency, value and graphs. International Journal of Game Theory, 49(1):227–249, 2020.

---

### Official Review · Reviewer_9p2U · 2022-07-10

**Rating:** 7
**Confidence:** 5
**Soundness:** 3 good
**Presentation:** 3 good
**Contribution:** 3 good

**Summary:**

The paper proposes a structure aware scoring function, referred to as HN value, to explain decisions of a GNN for the task of graph and node classification. HN value originally proposed in reference [6] of the paper has its roots in cooperative game theory like Shapley values but unlike Shapley values, it is structure aware.  An algorithm called GStarx is proposed which computes HN values corresponding to nodes in a graph later presented as explanations. Higher the HN value higher the node importance. The superiority of the method is demonstrated via experiments in 6 graph classification datasets.

**Questions:**

1. I am still wondering about the lack of feature importances in the explanation. For a graph we are given nodes, their features and the connections. The current paper address the problem of finding the most important nodes, which might be reasonable for the datasets used currently in the paper. But for real world datasets both for node and graph classification, node features may play an important role. How do or can we find feature importances together with node importances with the current methodology?

2. Definition 4.2 is a bit confusing.  It seems like the condition |S/G|=1 will only be satisfied if the induced graph on nodes in S is a clique.  But intuitively, the first condition should correspond to graph induced on S to be a connected subgraph. The authors should elaborate on definition 4.2 a bit more and justify the choice even if it was originally proposed in [6]. The example given in lines starting from 180 is too simple as it only considers the case when |S|=1.

3. In the GStarX algorithm, what is f^0_c^*?

4. It is not clear if the obtained explanations are actually sparse as the importance scores are continuous. For example, a method can lead to a uniform importance score distribution over the nodes, i.e., all nodes are almost equally important. One should use entropy based metric for sparsity as proposed in https://arxiv.org/abs/2105.08621  to show that indeed the importance score distribution is not very uniform.

One minor question: What does HN stand for? I even looked up in the original paper [6], could not find the full form!

**Limitations:**

One of the main limitations of this work is that there is no direct way to compute feature importances while explaining a decision which might be very important in some real world datasets and applications. Consequently in presence of feature explanations more evaluation metrics might be required as one can just not simply remove a feature while computing fidelity/faithfullness as argued in https://arxiv.org/abs/2105.08621

The authors should explicitly acknowledge this limitation while still providing examples of application areas for example for text graphs or graphs with dense feature representations extracted using some unsupervised representation learning methods. In those cases, providing additional feature explanations might not be useful as the features themselves are not directly interpretable.


**Strengths And Weaknesses:**

Strengths:

1. The paper proposes a new scoring function based on cooperative game theory to find node importances.
2. The paper is well written and structured.
3. The overall study is carried out well with various concerns, regarding application to node classification, comparison to closely related work based on C-Shapely, at least addressed in the Appendix.

Weaknesses:
1. The proposed method can only be applied to find node importances ignoring the differing feature importances. Note that a GNN uses both node features and structure to make predictions. Some of the features might be more important than others towards a decision.
2. There are a few clarifications required based on the current text which I also elaborate under Questions.

---

> ### Author Response · Authors · 2022-08-02
> **Response to Reviewer 9p2U (Part 2)**
>
> > Entropy-based sparsity evaluation as proposed in ZORRO [1]
>
> Thank you so much for pointing out this metric. We have followed Definition 2 in ZORRO and added evaluation results using the entropy-based sparsity. In particular, we computed the entropy of normalized scores GStarX output for each graph, and then averaged the entropy over each dataset. To understand how sparse these results are, we pick three distributions as references. 1) The entropy of uniform distribution over all n nodes in a graph, i.e., Uniform(n), which represents the least sparse output and is an upper bound of entropy-based sparsity as pointed out in the ZORRO paper. 2) The entropy of uniform distribution over the top 25% nodes in a graph, i.e., Uniform(0.25n), where probabilities of the bottom 75% nodes are set to zero. This case is very sparse since 75% of nodes are deterministically excluded, which can be treated as a practical lower bound of the entropy-based sparsity. 3) The entropy of Poisson distribution with mean 0.25n, i.e. Poisson(0.25n). This is a more realistic version of the sparse outputs in case 2). Instead of setting all 75% of nodes to have probability zero, we assume the probabilities for tail nodes decrease exponentially as a Poisson distribution while the mean is kept the same as in case 2).
>
> In the table below, we show the average entropy-based sparsity of GStarX outputs vs. these three reference cases on each dataset. We see that the GStarX sparsity is between Uniform(0.25n) and Uniform(n) and is close to Poisson(0.25n), which justifies that GStarX outputs are indeed sparse. We have included the Table and corresponding discussion in Section 5.2 in our revised PDF.
>
> |                    |   BA-2motifs  |   BACE    |   BBBP    |   SST     |   MUTAG   |   Twitter  |
> |--------------------|---------------|-----------|-----------|-----------|-----------|------------|
> |   GStarX scores    |   2.1352      |   2.4481  |   2.3290  |   2.3282  |   2.2434  |   2.2114   |
> |   Uniform(n)       |   3.2189      |   3.5080  |   3.0728  |   2.8698  |   2.8612  |   2.9833   |
> |   Uniform(0.25*n)  |   1.8326      |   2.1217  |   1.6893  |   1.4855  |   1.4749  |   1.5970   |
> |   Poisson(0.25*n)  |   2.3204      |   2.4686  |   2.2416  |   2.1336  |   2.1323  |   2.1945   |
>
>
> > What does HN stand for?
>
> Most of the cooperative game values are named after the inventors, e.g. the Shapley value, the Myerson value, the Hamiache value, etc. Since no formal name was given to the value proposed in Hamiache & Navarro 2020 [2], we follow this convention to refer to the value as the Hamiache-Navarro value (there is a Hamiache value already), and thus HN value for short. We have added this clarification in the introduction (line 44). Thank you for pointing it out.
>
> > Acknowledge the limitation of not explaining node feature importance.
>
> In the revised PDF, we have added a clarification in Section 4.3 (lines 194 - 197)  when we introduce the GStarX algorithm, which clarifies that our method doesn’t include node feature importance scoring. As we mentioned in our response to the first question, we also added in Section 7 that node feature importance explanation is a meaningful future work.
>
> Reference
>
> [1] Funke, Thorben, Megha Khosla, and Avishek Anand. "Zorro: Valid, sparse, and stable explanations in graph neural networks." arXiv preprint arXiv:2105.08621 (2021).
>
> [2] Gérard Hamiache and Florian Navarro. Associated consistency, value and graphs. International Journal of Game Theory, 49(1):227–249, 2020.
>
> [3] Lundberg, Scott M., and Su-In Lee. "A unified approach to interpreting model predictions." Advances in neural information processing systems 30 (2017).

---

> > ### Comment · Reviewer_9p2U · 2022-08-08
> > **Thank you for addressing the comments**
> >
> > The authors have addressed my comments.

---

> ### Author Response · Authors · 2022-08-02
> **Response to Reviewer 9p2U (Part 1)**
>
> We truly appreciate your comments and support for our paper! Your suggestions make our paper clearer and stronger. We hope our response can address your questions. Please find our detailed response below. The PDF is also revised accordingly with changes highlighted in red.
>
>
> > Feature importance is also critical and meaningful in GNN explanation. How to include it into GStarX?
>
> We totally agree with the reviewer that different features can have different importance for GNN predictions. Feature importance explanation is thus a meaningful and critical part of practical GNN explanation. This point has been studied together with node importance in the earlier GNN explanation works like GNNExplainer, as well as some recent milestone works like ZORRO [1] suggested by the reviewer. We choose to exclude the feature importance explanation for two reasons.
>
> * Feature dimensions of many graph datasets are not interpretable. Like the text graph dataset GraphSST2 used in SubgraphX and our paper, the node features are 768-dimensional word vectors. Identifying the importance of each dimension for these node features can thus be less meaningful. On the other hand, the text graphs are probably the best type of graphs for qualitative evaluation of explainers, given the lack of ground truth for explaining general graphs. The importance of keywords (nodes) to sentence sentiment prediction (graph label prediction) can be easily understood when visualized.
>
> * For many of the recent works like PGExplainer, SubgraphX, OrphicX, etc, the explanation was only performed at the node/edge level. Their argument is that feature importance is not unique to graphs and has been extensively studied in general model explanability literature. As a work focusing on GNN explanation, feature importance is good to have but may not be a core contribution. Given these methods are important baselines for our method, we choose not to include node feature importance at this time for a fair comparison to these methods.
>
> Nevertheless, feature importance explanation is indeed important and can be incorporated into our method as well. One idea is to perturb the graph nodes and node features together as in the GraphSVX paper. Specifically, when explaining a graph with N nodes and feature dimension D, instead of scoring induced subgraphs by picking nodes from the size-N set, we can extend the scoring function to take in an induced subgraph with a subset of features, i.e., picking from the size-(N+D) set. The unpicked features can be set as zero. In this way, the feature importance can be included to make the final explanation more complete. Given the limited time of the rebuttal period, we leave the implementation of this idea and the exploration of smarter ways for including feature importance as future work. We have revised the PDF to clearly stated our limitation of feature importance explanation in Section 4.3 (lines 194 - 197) and state this future direction in Section 7 (line 337 - 339).
>
> > Notation in Definition 4.2 is not clear.
>
> Thank you for pointing out this lack of clarity.  Your intuition is indeed correct; |S/G|=1 is satisfied if the graph is connected. In fact, we use the notation S/G to indicate node partition only to be consistent with the original HN value paper [2] and the convention of the game theory literature. S/G is intended to communicate a set of connected components for nodes in S. In the revised PDF, we have further clarified the definition of S and S/G in lines 155 - 159 to avoid confusion.
>
> > In the GStarX algorithm, what is $f^0_{c^*}$?
>
> $f^0_{c^*}$ is a normalization term used as a reference point. As we defined in lines 203,  $f^0_{c^*} = \mathbb{E} \left[ \left[ f(G) \right] _{c^*} \right]$.
>
> For this notation, the superscript $f^0$ stands for the expected prediction for an arbitrary graph in the dataset, and the subscript $f_{c^*}$ stands for the specific class the model predicted. We subtract it from $\left[ f(g_S) \right]_{c^*}$ so the explainer can identify negative contributions as well. Otherwise, all the scores will be positive since they come from probability outputs, and thus obscure negative contributions. Similar ideas have been adopted in Shapley-value-based works like SHAP [3].

---

### Official Review · Reviewer_Wfcz · 2022-07-11

**Rating:** 7
**Confidence:** 4
**Soundness:** 2 fair
**Presentation:** 2 fair
**Contribution:** 3 good

**Summary:**

- This paper proposes a new GNN explanation method called Graph Structure-aware eXplanation (GStarX) based on HN value from the cooperative game theory, and claims it is able to leverage the critical graph structure information and improve the GNN explanation.
-  This paper also tries to explain why and how this method is more effective compared with existing methods based on Shapley Value claimed to be non-structure-aware.
- Experiments conducted on chemical graph property prediction and text graph sentiment classification are used to demonstrate that GStarX produces qualitatively more intuitive explanations, and quantitatively improves explanation fidelity


**Questions:**

- Why the fidelity is used as the only metric for evaluation? How about metrics such as AUC and accuracy used in some other GNN explanation papers?
- Is the evaluation in Table 1 (with 8 different Sparsity) meaningful or fair?
- Algorithm 1 illustrates the main framework of the paper, but it seems Algorithm 3 is equally important for better understanding. It may be more appropriate to include Algorithm 3 in the main body of the paper.


**Limitations:**

There is no potential negative societal impact of this paper.

**Strengths And Weaknesses:**

- Strengths
    - This paper is first to apply HN-Value (borrowed from game theory) to GNN explanation and illustrates its effectiveness.
    - Compared with non-structure-aware GraphSVX, HN value utilizes graph structures to attribute cooperation surplus between neighbor nodes, so that node importance scores reflect not only the node feature importance, but also the structural roles.
    - SubgraphX is also structure-aware for GNN explanation, but it has limitation of only generating one single connected subgraph as explanation. GStarX can overcome the issue by generating multiple disconnected subgraphs as explanation.
- Weaknesses
    - The paper does not clearly explain how Equation 6 and 7 are transformed into the form of matrix multiplication in Algorithm 3, and some necessary explanations are needed for better understanding.
    - The paper claims that "Connecting GNNs and the HN surplus allocation through the message passing lens" and uses a molecule example shown in Figure 1 (c) to demonstrate  the connection between structure-aware HN value and the GNN message passing. A graph for sentiment classification in Figure 1(b) is used to qualitatively show that the structure-aware HN value can eliminate the inappropriate word coalition. But for better understanding, it seems there is lack of clear illustration using a unified explanation mechanism, which can naturally connecting these examples of message passing and elimination.
    - The example in Figure 1 (b) is only qualitatively studied, and for consistency and better illustration, it should be included in section 5 with the detailed quantitative analysis.

---

> ### Author Response · Authors · 2022-08-02
> **Response to Reviewer Wfcz (Part 2)**
>
> > AUC and accuracy as evaluation metrics
>
> To use metrics like AUC and accuracy, we need to know the ground truth explanation, i.e., labels saying which nodes/edges are important. For the earlier GNN explanation works like GNNExplainer, most of the datasets are synthetic and thus with ground truth available. For example, nodes in the motif graph in the BA2Motif dataset are treated as the ground truth while nodes in the BA graph are excluded from the ground truth. For those real datasets used more often in recent works like SubgraphX, we can’t apply AUC or accuracy because we don’t know the ground truth. Some strong baselines we compared to, e.g. the GraphSVX model, have achieved over 0.93 accuracy on all the synthetic datasets evaluated in the paper, but only achieve 0.77 accuracy on the real MUTAG dataset (among many existing works the only real dataset with ground truth available). We thus think that showing explainers can work well on real datasets is a much more challenging and meaningful task. Therefore, we selected real datasets for most of our experimentations, where we can only apply fidelity/inverse fidelity/sparsity for evaluation.
>
> > Is the evaluation in Table 1 (with 8 different Sparsity) meaningful or fair?
>
> Evaluation metrics fidelity/inverse fidelity/sparsity are adopted from [1,2,3]. For real datasets without the ground truth, fidelity and inverse fidelity provide meaningful evaluation. However, the difficulty is that the fidelity of explanation with different sparsity is not directly comparable. For example, using the whole graph as an explanation will most likely result in much higher fidelity than using any single node in a graph. However, the whole-graph explanation is not sparse at all, and directly comparing these two fidelities is not fair. SubgraphX chooses to compare methods using fidelity vs. sparsity plots for a range of sparsity. We have both the fidelity vs. sparsity plot and the inverse fidelity vs. sparsity plot for each dataset in Figure 4 in Appendix A.4. These plots show that our method outperforms baselines in most cases. The numbers in Table 1 are a normalized summary of the curves shown in Figure 4, it summarizes the fidelity and inverse fidelity analogously to the F1-score summarizes precision and recall, so we can quickly compare methods using a single number, which saves space for showing all the plots and save time for interpreting all the curves in the plots. Nevertheless, the plots are shown in Figure 4 in Appendix A.4 for a closer comparison.
>
> > Include Algorithm 3 in the main body of the paper
>
> Thank you for the suggestion. In the revised PDF, we have followed your suggestion to include Algorithm 3 (The Compute HN function) in the main body right next to Algorithm 1 at the top of page 6. Due to LaTeX labeling, it is shown as Algorithm 2 now.
>
> Reference
>
> [1] Pope, P. E., Kolouri, S., Rostami, M., Martin, C. E., and Hoffmann, H. Explainability methods for graph convolutional neural networks. In Proceedings of the IEEE Conference on Computer Vision and Pattern Recognition, pp. 10772–10781, 2019.
>
> [2] Hao Yuan, Haiyang Yu, Jie Wang, Kang Li, and Shuiwang Ji. On explainability of graph neural networks via subgraph explorations. In Marina Meila and Tong Zhang, editors, Proceedings of the 38th International Conference on Machine Learning, volume 139 of Proceedings of Machine Learning Research, pages 12241–12252. PMLR, 18–24 Jul 2021
>
> [3] Yuan, H., Yu, H., Gui, S., and Ji, S. Explainability in graph neural networks: A taxonomic survey. arXiv preprint arXiv:2012.15445, 2020c.

---

> > ### Comment · Reviewer_Wfcz · 2022-08-07
> > **Comments on the authors' response**
> >
> > Thanks for the detailed response. Generally, I think my concerns have been addressed and my questions have been answered.

---

> ### Author Response · Authors · 2022-08-02
> **Response to Reviewer Wfcz (Part 1)**
>
> We truly appreciate your comments and support for our paper! Your suggestions make our paper clearer and stronger. We hope our response can address your questions. Please find our detailed response below. The PDF is also revised accordingly with changes highlighted in red.
>
> > How are equations 6 and 7 transformed into the form of matrix multiplication in Algorithm 3?
>
> This transformation is the Lemma E.1 in Appendix E, which corresponds to the Lemma 1 in the original HN value paper [1]. This is a nontrivial lemma that takes two and a half pages to prove in [1] (page 230 - 232). In our opinion, although the math is elegant, the proof itself may be less interesting for many of the readers of our paper. Therefore, we choose to direct our readers to [1] when we mention our Lemma E.1 instead of rephrasing the proof in our paper. Our apologies for the inconvenience.
>
> > Figure 1(b) and 1(c) use different examples to illustrate elimination and connection to GNN message passing. A unified illustration can help for better understanding.
>
> We appreciate your suggestion for using a unified illustration. When making this figure, we intentionally picked three different types of graphs, i.e., a synthetic graph, a text graph, and a molecule graph. These three types correspond to the graph types we used in our experiment section, and they cover graph types that appeared in all of the existing GNN explanation methods we are aware of. Given the limited space of a paper, we hope a wide coverage of different examples can help our readers better understand our paper.
>
> Nevertheless, just as you pointed out, these two points can be illustrated using a unified example. For example, the text graph in Figure 1 (b) can also be used to demonstrate the connection between surplus allocation and GNN message passing. As we discussed in Section 4.2, HN surplus allocation and GNN message passing both essentially reflect the idea that close neighbors heavily influence each other (interpreted as “cooperation” in game theory, and “message passing” in GNNs), while far away nodes have a lesser influence on each other. In the text graph, this can be demonstrated via the phrase “not a bad”. It corresponds to a subgraph with three locally connected nodes, and these nodes should influence each other more than far away nodes. When considering the importance of the node “bad”, if all the coalitions it can form with any nodes in the graph are treated equally important, its sentiment contribution is likely to be negative. In contrast, if its role as part of the phrase “not a bad” is emphasized more than its other coalitions with far away nodes, its sentiment contribution can be less negative. As we respond in more detail to your next question, we have followed your suggestion to add a quantitative study of this example in Section 5.2. When comparing the normalized contribution score of the Shapley-based method GraphSVX and our HN-value-based method GStarX, we got the contribution of “bad” being -0.02242 and 0.00026 respectively. The contribution of “bad” for this sentence is turned from negative to neutral/slightly positive.
>
> > A quantitative study of the text graph in Figure 1(b)
>
> Thank you so much for the suggestion. Having a quantitative study of the example in Figure 1(b) definitely helps our paper to be more consistent. We have put it as the positive sentence in Figure 2, and moved the original positive sentence example in Figure 2 to the Appendix due to the space limit. The observation is similar to the original positive sentence in Figure 2. GStarX can select important sentiment words more accurately than baselines. We also add some discussion to echo the discussion of this example in Section 3.2, where we mentioned that the Shapley value will downgrade the positive importance of the word “good” for the upper sentence. Comparing the normalized contribution scores of our HN-value-based method GStarX and the Shapley-based method GraphSVX, “good” ’s contribution is higher in ours: 0.1152 vs. 0.0371. This discussion is now added in lines 268 - 271 and highlighted in red.

---

### Author Response · Authors · 2022-08-09
**General Response to All Reviewers**

We truly appreciate the constructive comments from all reviewers. We briefly summarize the major changes we made during the rebuttal period below. The paper PDF is also revised accordingly with changes highlighted in red.

* Add a detailed quantitative study of Figure 1(b) in Section 5.2
* Add entropy-based sparsity evaluation in Table 2
* Add discussion and potential ideas of including node feature importance into our framework in Section 4.3 and 7

As acknowledged by the reviewers, our major contributions include:

* Identify the non-structure-aware limitation of the Shapley value for GNN explanation
* Connect the structure-aware HN value to GNN message passing and GNN explanation
* A new structure-aware GNN explanation method outperforming baseline explainers

Regarding Reviewer ketX’s question about the original theoretical contribution, though there were no “theorem-proving” sections in our paper, our theoretical analysis of the Shapley value vs. the HN value in the GNN setting and the connection between the HN value and GNN message-passing are both our original contribution. The HN-value also cannot be applied to GNNs off-the-shelf, even theoretically. It requires a proper design of the characteristic function, Monte-Carlo approximation for scalability, etc, which are core problems we solve in our method. Finally, there are many more well-studied values in the rich game theory literature. Our work serves the purpose of highlighting this avenue for interplay between game theory and ML, and shifting focus beyond the Shapley value. Given this enlightenment to the community, we believe deeper theoretical studies from the ML perspective are soon to come.

We hope the changes and the discussion above further clarify our method and contribution. We thank all the reviewers again, and we are more than happy to discuss any other questions or comments the reviewers may have.

---

### Meta-Review · Area_Chair_j6GK · 2022-08-23

**Recommendation:** Accept
**Confidence:** Certain

**Metareview:**

The paper considered the task of identifying the most important subgraph and constituent nodes for graph-level predictions. It argued that the popular Shapley value is non-ideal since it's not structure-aware. It then proposed a structure-aware method, a scoring function based on the HN value from the cooperative game theory. Empirical results show that the method produces qualitatively and quantitatively improves over strong baselines.

The paper considers an important topic and has made novel and interesting contributions: the discussion that the Shapley value is not ideal for graph explanation, the non-trivial application of the HN value in this context, the well-design method and the thorough study showing its effectiveness. The authors have addressed well the comments by the reviewers.

**Award:**

No

---

### Decision · Program_Chairs · 2022-09-14

Accept